# Multifactorial Evaluation of Spatial Suitability and Economic Viability of Light Green Bridges Using Remote Sensing Data and Spatial Urban Planning Criteria

Hnin Wuit Yee Kyaw [1,*], Angeliki Chatzidimitriou [1], Jocelyne Hellwig [1], Michael Bühler [1], Johannes Hawlik [2] and Michael Herrmann [3]

[1]  Faculty of Civil Engineering, Konstanz University of Applied Sciences, 78462 Konstanz, Germany
[2]  Octopus Real Estate Development GmbH, 70619 Stuttgart, Germany
[3]  Str.ucture GmbH, 70176 Stuttgart, Germany
*  Correspondence: hnin.wuityeekyaw@htwg-konstanz.de; Tel.: +49-15730753792

**Abstract:** Multi-faceted stresses of social, environmental, and economic nature are increasingly challenging the existence and sustainability of our societies. Cities in particular are disproportionately threatened by global issues such as climate change, urbanization, population growth, air pollution, etc. In addition, urban space is often too limited to effectively develop sustainable, nature-based solutions while accommodating growing populations. This research aims to provide new methodologies by proposing lightweight green bridges in inner-city areas as an effective land value capture mechanism. Geometry analysis was performed using geospatial and remote sensing data to provide geometrically feasible locations of green bridges. A multi-criteria decision analysis was applied to identify suitable locations for green bridges investigating Central European urban centers with a focus on German cities as representative examples. A cost-benefit analysis was performed to assess the economic feasibility using a case study. The results of the geometry analysis identified 3249 locations that were geometrically feasible to implement a green bridge in German cities. The sample locations from the geometry analysis were proved to be validated for their implementation potential. Multi-criteria decision analysis was used to select 287 sites that fall under the highest suitable class based on several criteria. The cost-benefit analysis of the case study showed that the market value of the property alone can easily outweigh the capital and maintenance costs of a green bridge, while the indirect (monetary) benefits of the green space continue to increase the overall value of the green bridge property including its neighborhood over time. Hence, we strongly recommend light green bridges as financially sustainable and nature-based solutions in cities worldwide.

**Keywords:** green bridge; remote sensing; GIS; 3D urban planning; MCDA; cost-benefit analysis; financial sustainability; sustainable infrastructure; land value capture





## 1. Introduction

Substantial attention has been paid to urban sustainability as the majority of the world's population lives in cities [1]. Cities are interconnected and integrated in the global economy, which makes them intertwined in the risk landscape [2]. Despite being centers of opportunity, cities often disproportionately face environmental problems and climate impacts such as heatwaves, floods, and air pollution [3] due to their social, ecological, and infrastructural vulnerability. Sustainable measures like nature-based solutions (NbS) are urgently needed for vulnerable urban centers [4].

Nature-based solutions (NbS) are actions related to the protection, sustainable management, and restoration of ecosystems to address societal challenges [5]. On account of the multifunctional benefits of nature-based solutions, they are proliferating as an integrated approach that contributes to achieving all Sustainable Development Goals (SDGs) [6]. Recent developments of the evidence base in climate, social, and economic benefits of NbS,

attract the interest of urban planners and city governments to strengthen the resilience of cities [7]. An example can be seen in the EU research strategy [7]. Scholars have provided scientific evidence on the capacities of NbS on flood impact reduction [8–10] and air quality improvements [11].

However, implementation of NbS is usually hindered by competition from other land uses since NbS are place-based solutions. Strategies are required to deal with conflicts of interest such as competing land uses that emerge from the various goals and thoughts of landowners in cities [12]. In addition, utility service networks such as water pipes and electricity grid, and other critical infrastructure also reduce the priority of NbS implementation [13].

Moreover, the growing population, urban migration, and the need for associated services are additional reasons for space intensiveness in cities. On one hand, settlement extension stands as the major competing land use to the Nature-based solutions. Thus, the associated housing shortages require municipalities to focus on internal development and on urban redensification [14,15]. On the other hand, ongoing urbanization compromises ecosystems for conversion to urban sprawls [16], together with interrelated pressures such as ecosystem degradation and soil sealing [17,18]. Therefore, urbanization and associated demand for land urgently need new solutions to make space for nature-based solutions, housing shortages, and other important needs in the face of urban challenges.

Another common barrier to the implementation of NbS and real estate development in urban areas is related to financial aspects [6,19]. Long-term financial plans are required for the implementation phase, as well as for the maintenance part of NbS projects [6,20]. It is especially concerning when projects do not have enough funds to harness the long-term socioeconomic benefits of NbS [6]. This means that the economic aspects and financial security of NbS projects are essential in the decision-making to implement NbS projects, highlighting the fact that strong and long-term financial models can facilitate the decision-making and realization of NbS projects.

To facilitate the financial security of NbS projects, many other studies have developed different financial arrangements for funding NbS projects. For example, the incorporation of ecological indicators and natural capital stocks in fiscal transfers among city municipalities can support the mobilization of green infrastructure [13,19]. Public-private partnerships can flow the private capital to the NbS implementation and maintenance fostering risk and benefit sharing [19]. However, those approaches still do not solve the problem of space competition.

Therefore, to deal with land use competition, urbanization, and economic constraints for NbS, this paper will present a novel approach: lightweight green bridges as green infrastructure in the inner-city areas in Germany. These types of green bridges will include a combination of green spaces and real estate spaces. Therefore, according to the definition of NbS [5], through the development of green spaces, thus the creation of green infrastructure, green bridges can be regarded as nature-based solutions. These green bridges provide a complementary solution to redensification from a practical application perspective. Inner-city traffic lanes are to be spanned at particularly suitable sections with the aid of a lightweight construction ("green bridge") (see Section 1.1). Lightweight nature of the bridge will reduce costs of materials and resources used. Importantly, the hybrid combination of green spaces and real estate development of the green bridges will serve as major financial support for investing in urban green bridge development.

Looking at the state-of-the-art knowledge on the concept and research of urban lightweight green bridges in inner-city areas, it is still in its early stage of development. Green bridge research is mainly dominant in studies on conventional green bridges of wildlife crossings which are usually located away from urban centers [21,22]. Among those green bridge (wildlife crossing) studies, many studies also focused on optimal location searches for green bridges. For example, a study by Herkenrath (2022) focuses on finding optimal locations for wildlife crossings and green bridges to reconnect fragmented habitats of animal species [23]. Research by Fluschnik and Kellerhals (2021) emphasized addressing

the problems of placing wildlife crossings using multi-variate analysis [24]. Marzouk (2014) also developed a green-bridge rating system using Simo's procedure [25]. Some researchers use green bridges for traffic solving purposes [26]. However, those studies are not related to our concepts, types, and purposes of green bridges (see Section 1.1 for our concept of green bridges). Herrmann et al. (2019) developed a novel concept for lightweight green wildlife bridges, where they also mentioned the potential for inner-city bridges [27]. Only very limited research projects focused on similar types of green bridges. For example, Freiberg am Neckar and Hamburg with an idea of establishing green bridges with real estate development [28,29]. Moreover, the Zukunft Bau research project "über_dacht" of the University in Stuttgart, Institute for Living and Design (Institut Wohnen und Entwerfen: IWE) is doing research for locations search and doing case studies to generate living spaces over existing streets and motorways in Germany [30].

Therefore, we addressed this research gap with an innovative concept of the green bridge which can be upscaled by proving its geometrical, social, economic, and environmental suitability using geometry analysis, MCDA, and cost-benefit analysis. The green bridge will be a combination of real estate modules and green spaces to facilitate urbanization and NbS in a financially sustainable way. It corresponds to value creation, value enhancement through the upgrading of the adjacent development, and emission reduction. The basis for this research article was the research project "SuLiVaCo—Sustainable Lightweight Value Connect: Lightweight Green Bridges as a Key Element for Value Creation and Enhancement of Existing Urban Spaces", which was carried out as part of a 13-month lightweight construction innovation project funded by the Baden-Württemberg Ministry of Science, Research and the Arts. Our research will pursue the following research questions.

(1) Which locations can be geometrically feasible locations of green bridges considering the geometry and elevation of road networks of cities in Germany?
(2) Which geometrically feasible locations of green bridges can be spatially prioritized considering multi-dimensional factors to facilitate the effective implementation of green bridges?
(3) Is a selected suitable location (case study) of a green bridge economically viable?

Using a largely autonomous toolbox, the target technology enables the time-efficient analysis of a large number of potential areas. Within the scope of the final evaluation and verification, the exemplary implementation for representative selected areas was carried out, whereby a degree of concretization and differentiation was achieved (see Munich case study in Section 2.3.6), which served as a basis for the detailed planning at the identified suitable locations starting from this initial case study. By dissemination of the research results together with relevant industry stakeholders, the appropriate introduction of the research results into the field can be ensured.

### 1.1. Concept of Green Bridge

A green bridge proposed here is defined as a lightweight green bridge, which spans over inner-cities traffic lanes (Figure 1). The green bridge is a hybrid lightweight combination of residential buildings and public green areas. The bridge is composed of three segments (modules): M1 module-green bridge segment, M2 module-residential segment, M3 module-access bridge segment [28]. The covering is to be constructed over a tunnel using the cut-and-cover method [28]. The lightweight nature of green bridges is different from conventional green bridges. Importantly, the bridge is different from normal wildlife crossings in the fact that its purpose is to create new space for residential and green areas with a focus on inner-city parts.

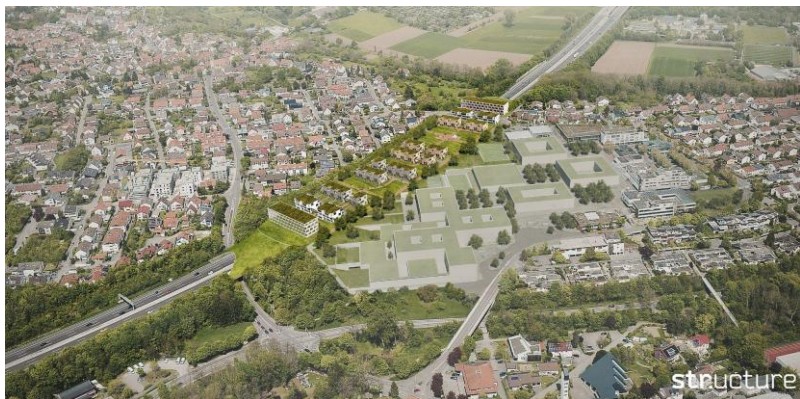

**Figure 1.** A green bridge in Freiberg am Neckar of the project "Freiraum für Freiberg". Photo from Str.ucture GmbH, 2019 [28].

## 2. Materials and Methods

### 2.1. Geometry Analysis Toolbox

This geometry analysis is aligned with the first research question and work package 1 of the associated project. This introductory work is to make sure necessary and practical foundations to determine the implementation potential of a green bridge at a given location. This geometry analysis will identify locations suitable for lightweight constructions ("green bridges") for spanning inner-city traffic lanes. Therefore, from a practical perspective, it is necessary to identify open tunnels which have sufficient depth from the neighboring structures. Sufficient depth means the appropriate height required to span lightweight structures and to continue the function of a tunnel as a road. An automated GIS model was built to perform the following necessary geometry processes as a toolbox for the selection of practical locations using the model builder tool in ArcGIS. The determination of the parameters and the considerations of geometry and heights were conducted in close negotiation with the engineering firms involved in the project. The processes included in the toolbox are described in Figure 2 and are explained in detail in the following sections. Processes were iterated for all the roads and tunnels in Germany using the toolbox.

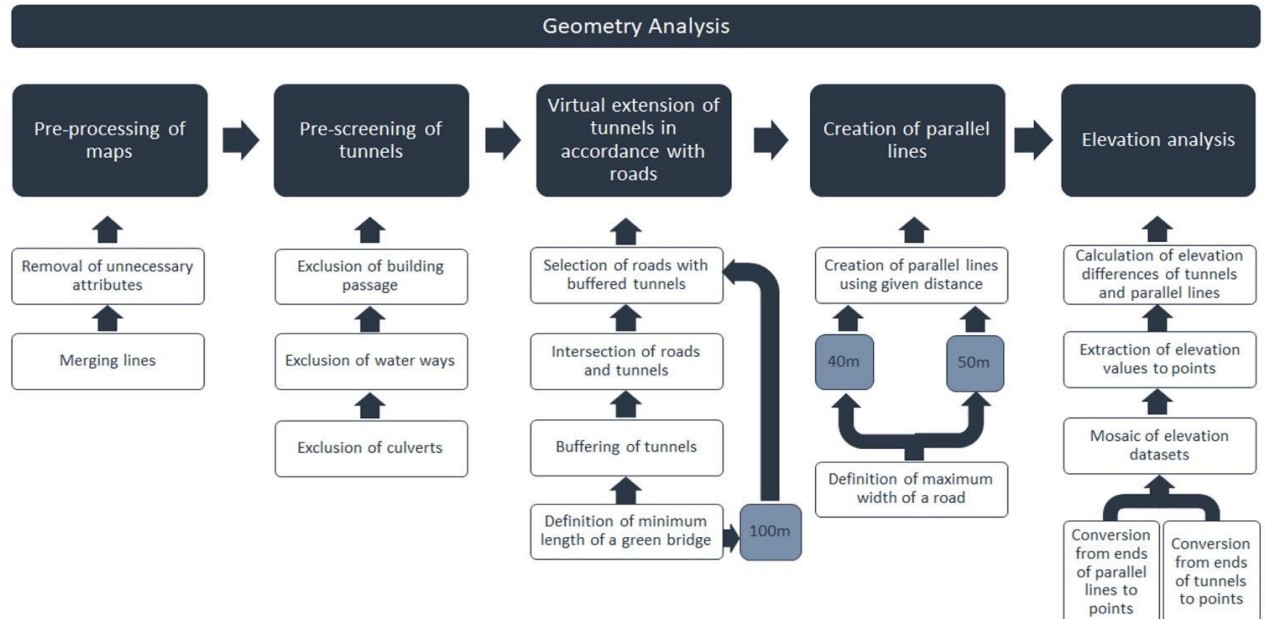

**Figure 2.** Processes of geometry analysis toolbox.

### 2.1.1. Pre-Screening of Tunnels

Since the idea of a green bridge is to span over an open tunnel, tunnels are used as a base element for the geometry analysis. They were screened with some fundamental criteria. The exclusion criteria used in the pre-screening stage were (1) tunnels that use as waterways, (2) tunnels under the buildings, and (3) culverts. Waterways such as ditches and building passages are not relevant for the purpose of a green bridge that uses open tunnels.

First and foremost, pre-processing of road and tunnel datasets was carried out. All the roads and tunnels were downloaded from the OpenStreetMap using polygons of counties of Germany. All the roads from the counties are merged and divided into 7 parts for the whole of Germany for the ease of processing large datasets. The division follows the alphabetical ascending order. All counties in each part were merged separately. Then, unnecessary attributes of the merged lines were removed using QGIS to reduce file size. Then, roads and tunnels were classified from the merged lines. This was followed by the extraction of tunnels from the road datasets and filtering with the exclusion criteria mentioned above.

### 2.1.2. Virtual Extension of Tunnels

After screening the waterways and building passages, the first geometry process was to virtually extend a closed tunnel following the road to a pre-defined distance. We extended both sides of the tunnel along the road to 100 m (Figure 3). 100 m is used as a pre-defined distance because the minimum length of a green bridge we proposed has a minimum length of 200 m. The extension process started with buffering the tunnels with the pre-defined distance. Then, it is followed by selecting roads and tunnels by locations using "intersect" and "shared a line segment with" relationships. The purpose is to extend the tunnels that follow the geometry of the roads rather than extending a straight tunnel from its ends. Then, the next steps were dissolving, spatial joining, and intersecting the roads with the buffered tunnels. The resulting extended lines were used to produce their endpoints using the "feature vertices to points" tool.

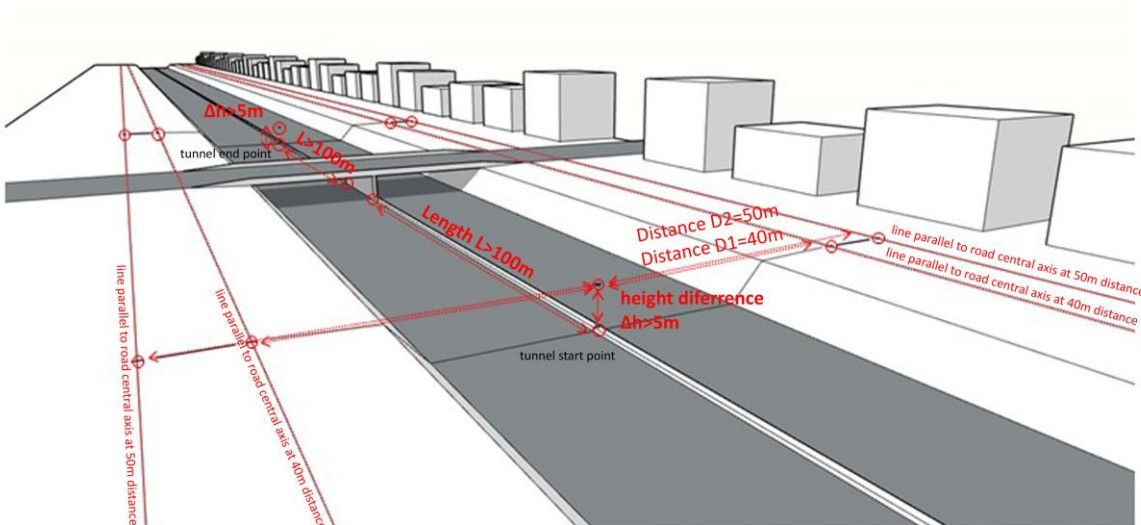

**Figure 3.** Distance measurement indication in the geometry analysis process.

### 2.1.3. Creation of Parallel Lines

The next geometry process was to develop lines parallel to the tunnels with a given distance. The purpose is to extract corresponding parallel points of the extended tunnels on the neighboring ground. The distance we used to create parallel lines is 40 m because the maximum width of a road in Germany is 43.5 m according to guidelines for the design of motorways by the Road and Transportation Research Association [31]. Considering the

two-way road systems, a 40 m distance of parallel lines is used to cover the small roads and 50 m to cover large roads. The combined results from both parameters are then intersected with each other to remove the overlapping locations.

Before the process started, the coordinate system of the project work was set to ETRS_1989_UTM_Zone_32N for Germany to facilitate spatial calculations. Then, lines parallel to the tunnels were created using a 40 m distance using ArcPy. After that, points were extracted from the ends of the lines as start and end points using the "feature vertices to point" tool. Then, it was necessary to reclassify as left and right lines of tunnels by using the "select by attribute" tool.

### 2.1.4. Elevation Analysis

The 3D aspect of the geometry analysis was reflected for the practicability of the locations of green bridges. The purpose of this analysis is to figure out if the extended ends of the tunnels are high enough to be able to implement a green bridge. The analysis can be divided into two parts: extraction of elevation by using points from the above processes and calculation of elevation differences between the extended tunnel points and their parallel points.

Reliable and high-resolution elevation datasets were required for this analysis to deal with roads and tunnels in an urban context. First, we started with freely available 30 m SRTM elevation dataset and 25 m EU Digital Elevation Model dataset from the Copernicus service. After validating the data with in-situ ground locations and Google Earth Pro, we found out that those datasets were not accurate enough to make elevation differences based on a given spatial difference between extended tunnel points and parallel points. Finally, we order a 5 m Digital Terrain Model dataset from the German Ministry of Cartography and Geodesy for the whole scale of Germany (https://www.bkg.bund.de/, accessed on 15 September 2022). The resulting data were validated to be reliable.

After the acquisition of the elevation dataset, the raster elevation ASCII files were mosaicked to be a single dataset to undergo the elevation analysis. Then, extended tunnel points and their parallel points from both sides were applied to extract their elevation values from the mosaicked dataset using the "extract multi-values to points" tool. The resulting points with their associated attributes of elevation values are joined to a middle point (i.e., extended tunnel points) to calculate elevation differences between them. Elevation values are subtracted from left points to middle points and from right points to middle points. Finally, a selection is made by querying to select the points of greater height than 5 m for the left and right differences to the middle points (Figure 3). 5 m is defined because it is the minimum height to build lightweight green bridge structures over the tunnels while allowing the function of the tunnels as roads. Minimum height is defined by the engineers in the research team. The sample resulting locations were validated using Google Earth Pro for their geometrical feasibility and are presented in Section 3.

### 2.2. Multi-Criteria Decision-Making Analysis (MCDA)

After selecting the potential locations where a green bridge is practically able to be implemented, we used a set of multi-dimensional criteria to develop an in-depth filter of the selected points from the geometry analysis (Figure 4). This in-depth multifactor evaluation allowed a pre-selection of short-listed sites for green bridges. This multi-criteria analysis was performed to answer the second research question of the paper and to fulfill work package 2 of the associated project. MCDA also serves as the benchmark tool to perform in-depth screening analysis. Criteria and their directions were defined based on a literature review of the related work and a survey on expert opinion from 35 experts (Table 1).

**Table 1.** Criteria used in the multi-criteria decision-making analysis for spatial suitability of green bridges.

| Criteria Group | Indicators | Direction | Aggregated Relative Weights (35 Experts) | Reference | Data Source |
|---|---|---|---|---|---|
| Ecological/environmental criteria | Extent of green space areas | "−" | 9.15 | [32–34] | [35] |
| | Amount of air pollutants (PM2.5) | "+" | 8.31 | [36] | [37] |
| Land use criteria | Extent of residential areas | "+" | 8.09 | [32] | [35] |
| | Extent of industrial areas | "-" | 6.12 | [32] | [35] |
| Economic criteria | Amount of land values | "+" | - | [36] | Not available |
| | Amount of building prices | "+" | - | [36] | Not available |
| | Average monthly income | "+" | 5.94 | [36] | [38] |
| Climate impact criteria | Intensity of urban heat islands areas | "+" | 8.63 | [33,34,36,39] | [40] |
| | Depth of flood hazard areas | "+" | 7.43 | [33] | [41] |
| Critical infrastructure criteria | Density of hospitals | "+" | 8.45 | [39] | [35] |
| | Density of schools | "+" | 5.91 | [39] | [35] |
| | Density of kindergartens | "+" | 5.81 | [39] | [35] |
| | Density of markets | "+" | 5.11 | [39] | [35] |
| | Density of power stations | "+" | 4.25 | [39] | [35] |
| | Density of transport stations | "+" | 6.64 | [39] | [35] |
| Social Criteria | Population density | "+" | 8.24 | [33,34,42] | [43] |
| | Density of elderly people | "+" | 5.10 | - | [38] |

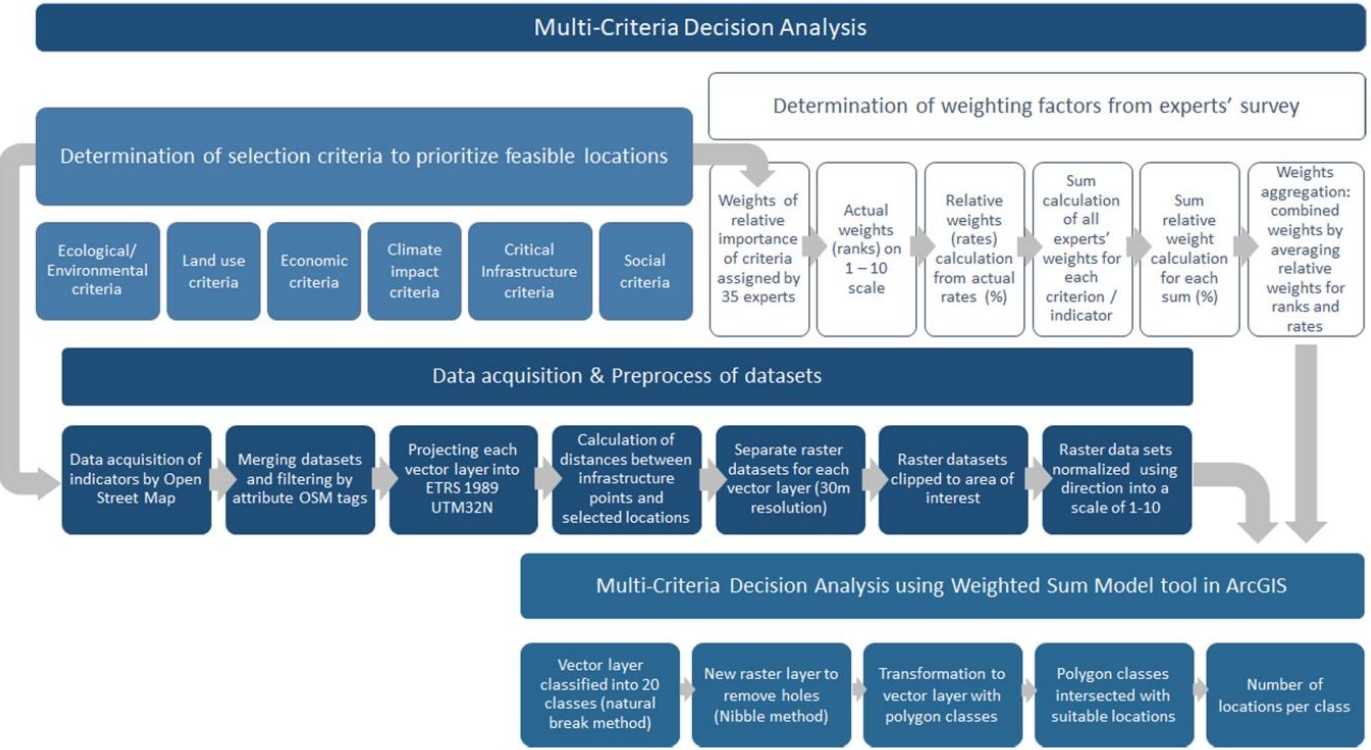

**Figure 4.** Process of the multi-criteria decision analysis.

### 2.2.1. Description of Criteria

As an in-depth analysis, multiple criteria are used to spatially prioritize geometrically feasible locations of green bridges in cities in Germany. These criteria can be grouped into ecological/environmental, economic, social, climate impact, and social/critical infrastructure criteria. Ecological criteria such as the extent of green space are used to understand the deficit of green space and associated ecosystem services [32,34]. Air quality is used to specify areas with high air pollution rates that need green spaces [36]. Land use considerations, such as residential and industrial areas, are identified to determine the spatial suitability of green bridges to implement in different land use systems. Economic criteria such as land values and building prices are used to enhance cost-effectiveness of green bridges in terms of providing revenue from real-estate modules. Average annual income was used as an indicator for understanding the economic status of populations. The positive direction was used for the average income criteria. However, it does not mean that richer areas have high priority. The justification is since the dataset has a scale of counties, we use rich cities that are capital cities in states such as Munich, Berlin, Stuttgart, and Cologne to be selected as they are the places where large diversity of economic classes exists.

Urban heat island criteria are used to identify areas which required green space to provide cooling functions [34]. We used yearly average of the summer months from the dataset of [40] accessing through Google Earth Engine platform. Flood extent is used to identify areas in need of water infiltration and de-sealed areas. Flood hazard depth and extent of 100-year-return periods were used as a climate impact factor to flooding risk. The spatial density of critical infrastructure is used for accessibility and protection of that infrastructure from climate impacts. For example, nearness to critical infrastructure such as train stations, schools, kindergartens, and markets provide greater access for real estate property on the green bridges. At the same time, the protection of that critical infrastructure including hospitals, markets, kindergartens, and power stations from heat and flood impacts is needed by the regulation function of green spaces. The collapse of that infrastructure may have a wider impact on socioeconomic systems, thus, undermining the sustainability of cities. Population density is used as a proxy indicator for the potential demand for space and ecosystem services [42]. The density of elderly people is used to spatially prioritize access to green infrastructure for vulnerable elderly people. This makes the decision-making process of spatial prioritization of green bridges practiced with a lens of equity and justice.

### 2.2.2. Weighting of Criteria

Weights are applied using expert opinions. A survey was conducted among the experts. A total of 35 experts participated in the survey to assign weights and relative importance of the criteria. Weights were collected according to the method of weighted sum model. The scale of weights ranged from 0 to 10. The resulted weights were aggregated using the guidelines for applying multi-criteria analysis to the assessment of criteria and indicators [38]. Firstly, relative weights were calculated. In other words, they are converted into percentages. They are called rates and the actual weights are called ranks. Sums of the rates and ranks were calculated across experts for each criterion. Moreover, relative weights for each sum were calculated by converting them into percentages. Finally, relative weights for ranks and rates were averaged to have combined weights.

### 2.2.3. Pre-Processing of Criteria

The MCDA analysis starts with data acquisition of the indicators for the national scale of Germany. After that, datasets were preprocessed to be the same raster data format. Preprocessing includes various stages depending on the dataset. Data that were downloaded from OpenStreetMap (OSM) underwent the process of merging and filtering using "select by attribute" tool. OSM tags that are used to define green spaces, residential areas, industrial areas, schools, hospitals, kindergartens, markets, wastewater plants, electricity stations, and transport stations were described in Supplementary Materials S1 (Tables S1–S9). Each

of the vector layers was then projected into The European Terrestrial Reference System Universal Transverse Mercator Zone 32N (ETRS 1989 UTM 32N). The distance between infrastructure points/land use areas and selected locations was calculated by using their Euclidean distance. This results in their respective raster datasets. A resolution of 30 m was used to be consistent for all the layers. The resulting raster datasets are then clipped to the extent of interest.

After preprocessing, each raster dataset was normalized using the directions described in the Table 1 into a scale of 1 to 100. Then, MCDA was performed using Weighted Sum Model tool in ArcGIS Pro. Weights described in the table were applied. The resulting vector layer was classified into 20 classes using natural break method. The natural break method was used to understand the original breaks of classes in the suitability scores. Then, the resulted raster layer was converted to a new raster using Nibble methods in order to make sure to remove any holes in the layer. Then, the raster layer with classes was transformed into a vector layer. Finally, the resulted polygon classes were intersected with suitable locations datasets to drive the number of locations per each class.

### 2.3. Cost-Benefit Analysis

#### 2.3.1. Cost Model

The cost model (Figure 5) mainly examines the cost aspect. The production costs are determined according to DIN 276, as recommended in the RBBau [44]. The calculation of the manufacturing costs is divided into three components [28].

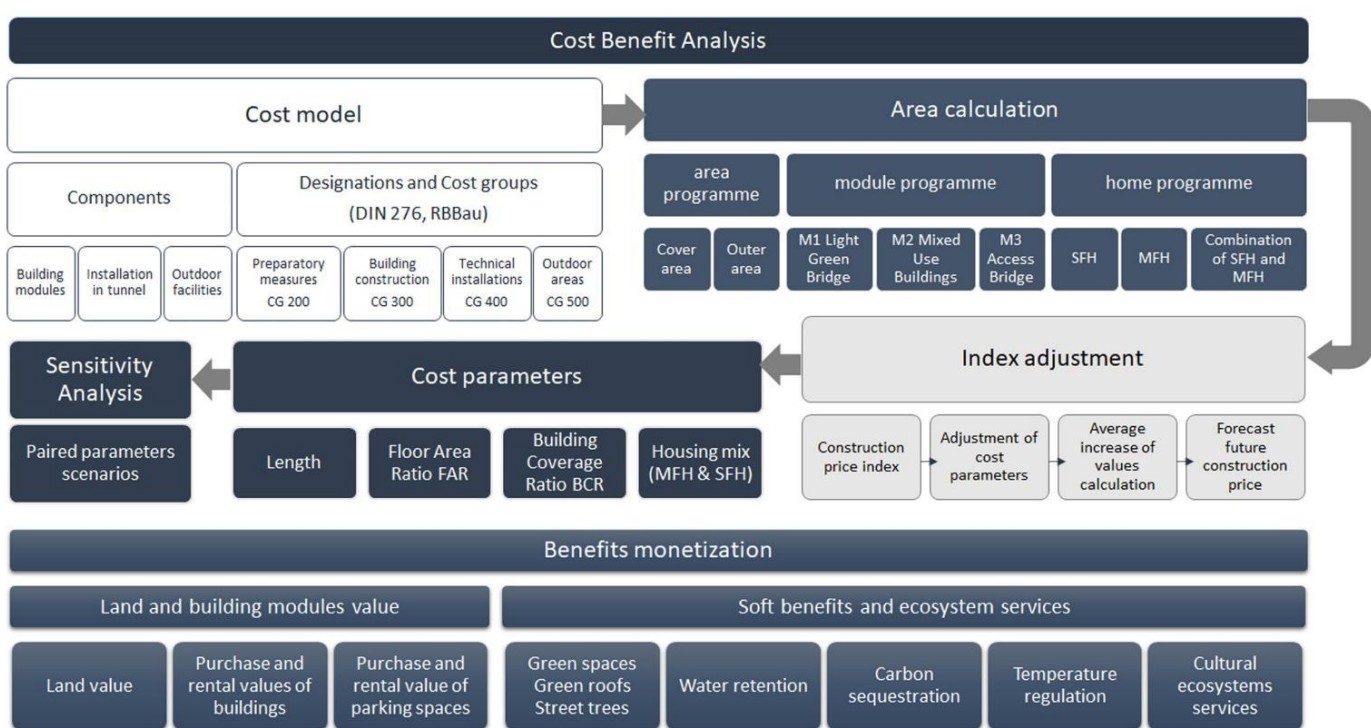

**Figure 5.** Process of the cost-benefit analysis.

Firstly, the costs of the cover which include building modules, installation in the tunnel, and outdoor facilities are calculated according to the cost group (CG) in Table 2. Then, the costs of the basement and building development are calculated with respective CGs.

**Table 2.** Classification of the calculation.

| Component | Designation | Cost Group |
|---|---|---|
| 1. Cover | Preparatory measures | CG 200 |
| | Building—cover construction, modules | CG 300 |
| | Technical installation tunnel | CG 400 |
| | Outdoor area next to cover (OA1) | CG 500 |
| | Outdoor area on cover, without M1 (OA2) | CG 500 |
| 2. Basement development | Basement development | CG 300–400 |
| 3. Building development | Single family household—living | CG 300–400 |
| | Multifamily household—mixed use | CG 300–400 |

### 2.3.2. Area Calculations

The next step was the area calculation of a green bridge. Area calculations are made in the three different levels. Level 1 contains the area program, which includes the outdoor areas. Here, the given width of the standard cross-section of 43.5 m is multiplied by the selected length of the project. This results in the cover area [31]. The same is done with the given width of the outer area next to the cover (OA1). The area OA1 was assumed to be a 16.6 m wide strip to the left and right of the cover [31]. This represents an average value. The cover area and OA1 add up to the total area of the project.

The next level is the module program. From this, not only the number of respective modules is determined, but also other area parameters, such as the gross floor area (GFA) of the development, the development area in the basement, and the external areas on and next to the cover with development.

The three modules M1—light green bridge, M2—residential building, and M3—access bridge can be used individually over the length of the project. Thus, there is no predetermined ratio structure. However, the model assumes that the permissible building coverage ratio (BCR) is fully utilized and this determines the number of M2 modules. The detailed description of area calculation of the modules and their arrangement is explained in Supplementary Materials S2.

In the final level, home program is discussed. The permissible GFA results from multiplying the total area by the GFA of the development plan of the surrounding properties. Depending on the planned proportion of single-family (SFH) or multi-family homes (MFH) in the project, the GFAs of the respective house types are formed. For example, a house mix with a 50% share of the respective house types also results in the division of the permissible GFA into 50% GFA for single-family houses and 50% GFA for multiple-family houses.

### 2.3.3. Index Adjustment

In order to adjust the cost parameters (CP) from the book of the Baukosteninformationszentrum Deutscher Architektenkammern (BKI) GmbH [45] to the current quarter, the construction price index was used, and the index adjustment of the CPs was carried out. A forecast was made for the third quarter of the year 2022. The construction price index for the second quarter of the year 2022, is +147.2% for residential buildings and +149.2% for office buildings according to the Federal Statistical Office [46]. The average value of the increases of the last quarter from the first quarter of 2021 was calculated for the forecast. This average increase value of +5.27% (residential buildings) and +5.53% (office buildings) is used for the forecast of the construction price index for the third quarter of 2022. As a result, it is assumed that by the third quarter of 2022, a construction price index of 152.5%

(residential buildings) and 154.7% (office buildings) can be expected compared to the base year 2015 [46]. All costs include VAT at the rate of 19% and 4.3 is indexed to the third quarter of 2022. The index adjustment for the cost parameters from the study of (Herrmann et al. 2020) from Q4 2019 to Q3 2022 is +1.32% [28]. The index for the cost parameters from the BKI (2022) from Q1 2022 to Q3 2022 is +1.10% in the residential sector and +1.11% in the office sector.

### 2.3.4. Cost Parameters

Three main parameters are expected to drive the overall cost of the project: the length, floor-area ratio (FAR) ratio, and the housing mix. While a higher FAR would increase the usable area, it is not expected to increase the cost of the modules. The average rates in horizontal and vertical series should be the same. The FAR is expected to be inversely proportional to the cost: the larger the FAR, the lower the cost expected. However, an expanding length of the bridge would increase costs significantly. The housing mix is also a driving factor in overall cost, with multifamily housing costing significantly more than single family housing. Detailed explanation of costs according to the cost parameters are described in Supplementary Materials S3.

### 2.3.5. Sensitivity Analysis

The sensitivity analysis determines how the costs behave when the parameters are changed. The parameters that can be changed are the length of the coverage, the BCR and FAR, and the house mix, or the proportion of single-family homes and multi-family homes. Three programs were developed and examined for analysis: the area program, the module program, and the housing program using different pairs of parameters. Different cost parameters were paired up to create scenarios and test their influence on the total costs. Those combination scenarios are described Table 3 and Figure 6.

**Table 3.** Scenarios of pairs of cost parameters for sensitivity analysis.

| Scenario | Length 200 m | Length 400 m | Length 600 m | Low Building Density BCR 0.2 FAR 0.4 | Medium Building Density BCR 0.5 FAR 1.7 | High Building Density BCR 0.8 FAR 3.0 | Only SFH SFH 100% MFH 0% | Mix $\frac{1}{2}$, $\frac{1}{2}$ SFH 50% MFH 50% | Only MFH SFH 0% MFH 100% |
|---|---|---|---|---|---|---|---|---|---|
| 2.3.1 | | x | | | | x | x | | |
| 1.3.1 | x | | | | | x | x | | |
| 3.3.1 | | | x | | | x | x | | |
| 2.3.2 | | x | | | | x | | x | |
| 1.3.2 | x | | | | | x | | x | |
| 3.3.2 | | | x | | | x | | x | |
| 2.3.3 | | x | | | | x | | | x |
| 1.3.3 | x | | | | | x | | | x |
| 3.3.3 | | | x | | | x | | | x |
| 3.2.1 | | | x | | x | | x | | |
| 2.2.1 | | x | | | x | | x | | |
| 1.2.1 | x | | | | x | | x | | |
| 3.3.3 | | | x | | x | | | x | |
| 2.2.2 | | x | | | x | | | x | |
| 1.2.2 | x | | | | x | | | x | |
| 3.2.3 | | | x | | x | | | | x |

**Table 3.** *Cont.*

| Scenario | Length 200 m | Length 400 m | Length 600 m | Low Building Density BCR 0.2 FAR 0.4 | Medium Building Density BCR 0.5 FAR 1.7 | High Building Density BCR 0.8 FAR 3.0 | Only SFH SFH 100% MFH 0% | Mix $\frac{1}{2}$, $\frac{1}{2}$ SFH 50% MFH 50% | Only MFH SFH 0% MFH 100% |
|---|---|---|---|---|---|---|---|---|---|
| **2.2.3** | | x | | | x | | | | x |
| **1.2.3** | x | | | | x | | | | x |
| **3.1.1** | | | x | x | | | x | | |
| **2.1.1** | | x | | x | | | x | | |
| **1.1.1** | x | | | x | | | x | | |
| **3.1.2** | | | x | x | | | | x | |
| **2.1.2** | | x | | x | | | | x | |
| **1.1.2** | x | | | x | | | | x | |
| **3.1.3** | | | x | x | | | | | x |
| **2.1.3** | | x | | x | | | | | x |
| **1.1.3** | x | | | x | | | | | x |

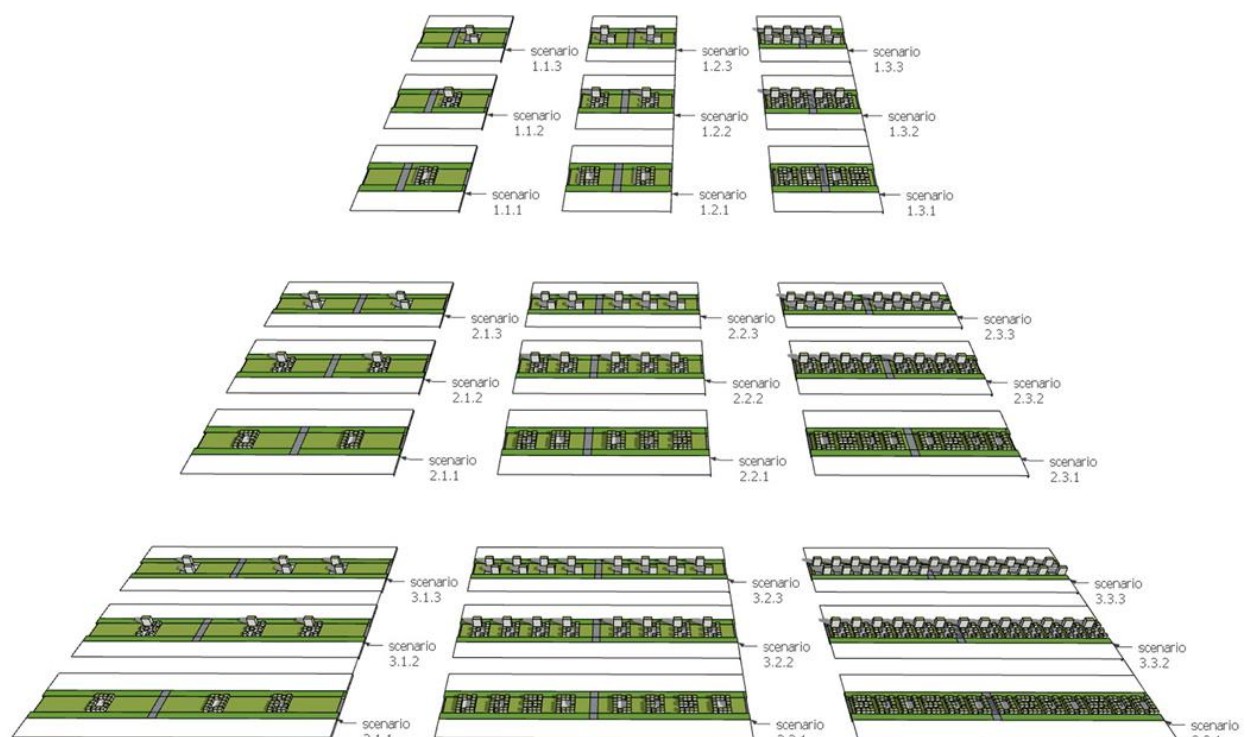

**Figure 6.** Visualization of different pairs of built-defnsity scenarios (sensitivity analysis).

2.3.6. Case Study McGraw Trench in Munich

The developed calculation model is not only used to analyze the parameters, but also for specific practical applications. In order to test its practical suitability, a location in Munich is examined as a case study since it is also selected as a study location at the res A cost-benefit analysis including the consideration of revenues is carried out, analyzed, and evaluated. It is a case study location selected in the Zukunft Bau research project "über_dacht" [30] and has been evaluated with the developed tools.

The location of the case study is in the southern part of Munich, in the district of Obergiesing. The "McGraw-Graben" runs between the St.-Quirin- Platz intersection to the

north and the southern overpass of Stadelheimer Straße. The section is 13pprox.. 450 m long, four-lane road constructed as an open tunnel.

2.3.7. Modification of the Model to the Case Study

Modifications to the calculation of costs and benefits are required to adjust to the conditions of the case study (Table 4). The length and width of the road as well as the width next to the cover were taken from Geographic Information System (GIS). The BCR, FAR, as well as the maximum number of stories, are based on the key data resolution of the urban development competition for the McGraw Barracks site in 2018. The house mix was adapted to the surrounding development, according to which only MFH is to be considered.

**Table 4.** The modified values that are considered in the calculation of the case study of McGraw Trench Munich.

| Parameters | Consideration of Revenue |
| --- | --- |
| Length | Approx. 450 m |
| Width (SC) | Approx. 20 m |
| BCR | 0.3 |
| FAR | 1.6 |
| House mix | 100% MFH, 0% SFH |
| Standard ground value | 2900 €/m$^2$ |
| Property interest | 5% |
| Capitalization factor | 19.34 |
| Discount factor | 0.0329 |
| Purchase price parking space | 35,000 €/pc. |
| Purchase price flat | 8849 €/m$^2$ |
| Rental price parking space | 100 €/m$^2$ |
| Flat rent | 17.39 €/m$^2$ |

The width next to the cover (OA1) is expected to be approximately 10 m with 5 m each on both sides. GFA is expected to be 914 m$^2$ due to the changes in width. BCR on the cover will be 100% while BCR next to the cover will be 0% as the width next to the cover is very narrow. Thus, 80.4% of the area of the M2 module is built on the cover. The M3 module was not applicable as the area is already bridged by several bridges that can be used.

For the benefits of the green bridge, so far, standard land value, and the purchase and rental prices of the building modules are considered. The average standard land value in this location is based on the inquiry to the real estate agency Fischer Immobilien in Munich. The property interest rate was assumed to be 5%. A capitalization factor is required to capitalize the net income and a so-called discount factor is required to add the discounted land value, which is formed from the property interest rate and the remaining useful life. The purchase and rental prices of the underground parking spaces and the average net cold rent are based on current values from the Immoscout24 internet portal [47,48].

Benefits provided from ecosystem services of a green bridge are calculated using the scenario provided in Table 5. The green space ratio was derived from the area of the M1 module adjusted for the case study. Since the residential property M2 module will have green roofs and photovoltaic panels on their roofs, the ratio of the green roof was estimated to be 80%. All the calculations were made using the assessment tool provided by the Institute for Ecological Economy Research (IÖW) [49]. The benefits that are included in the analysis were water retention, temperature regulation, carbon sequestration, and cultural ecosystem services.

**Table 5.** Calculation of soft benefits of green space in the green bridge.

| Criteria | Value |
|---|---|
| Total area | 9000 m$^2$ |
| Green space ratio | 44% |
| Green space area | 3960 m$^2$ |
| Green roof ratio | 80% |
| Street trees | 4 trees per 100 m |
| Green path | 30% |
| Maintenance of green space | 45% |

## 3. Results

### 3.1. Geometry Analysis

The first objective of the paper is to identify geometrically feasible locations for lightweight green bridges to be implemented in German cities in practice. Geometry analysis ensures this objective, and its results show a total of 3249 locations to be geometrically suitable to implement a green bridge (Figure 1). This means these locations have an open tunnel that has adequate depth/ height for a green bridge to span over. These locations can be distinguished to one side of the tunnels and the other sides of the tunnels as 1578 and 1671 locations, respectively. It is important to note that the locations of the two sides of each tunnel are not necessarily close to each other and their distance depends on the length of the respective tunnels.

The geometrically suitable locations are highly concentrated in the middle and southern cities of Germany and sparsely distributed in the north, except for big cities like Berlin and Hamburg (Figure 7). It can be clearly seen that the potential locations for a green bridge are densely located in the northwest areas of Germany, namely in the state of north Rhein-Westfalen. It appears that the potential locations followed the pattern of lines that go in the direction of north to south, following the existing roads and tunnels. This aspect is important for the connectivity of green spaces.

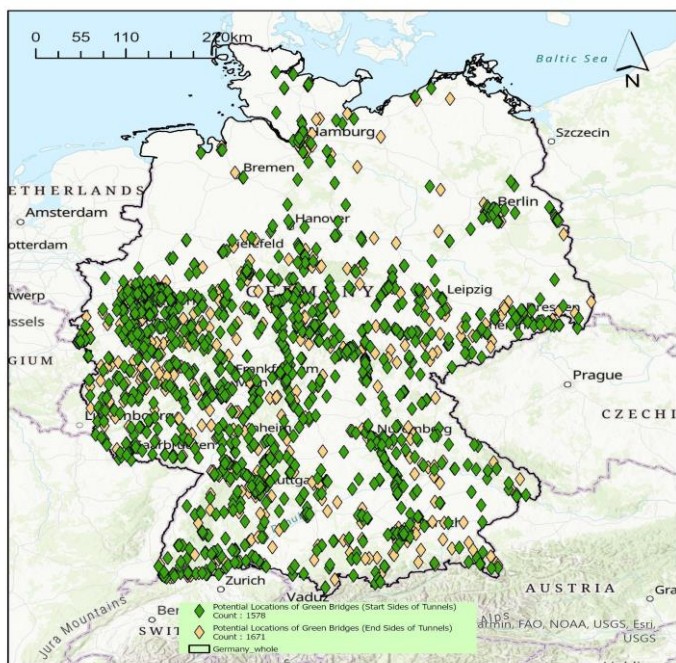

**Figure 7.** Potential locations of Green bridges in Germany.

Figure 8 presents certain cities with a relatively large number of geometrically feasible locations. At the provided zoom level as in Figure 8, the locations of the start sides and end sides seem to overlap with each other. Results show that when one side of a tunnel has a geometrical feasibility to implement a green bridge does not mean that the other side has it too. Some locations are concentrated, and some are dispersed, nonetheless, most of them occur in the inner city areas. Results show that Stuttgart and Berlin are likely to be the cities with the highest number of potentially suitable locations. However, it is difficult to infer their implementation potential at this stage since many factors are required to consider (See Section 3.2).

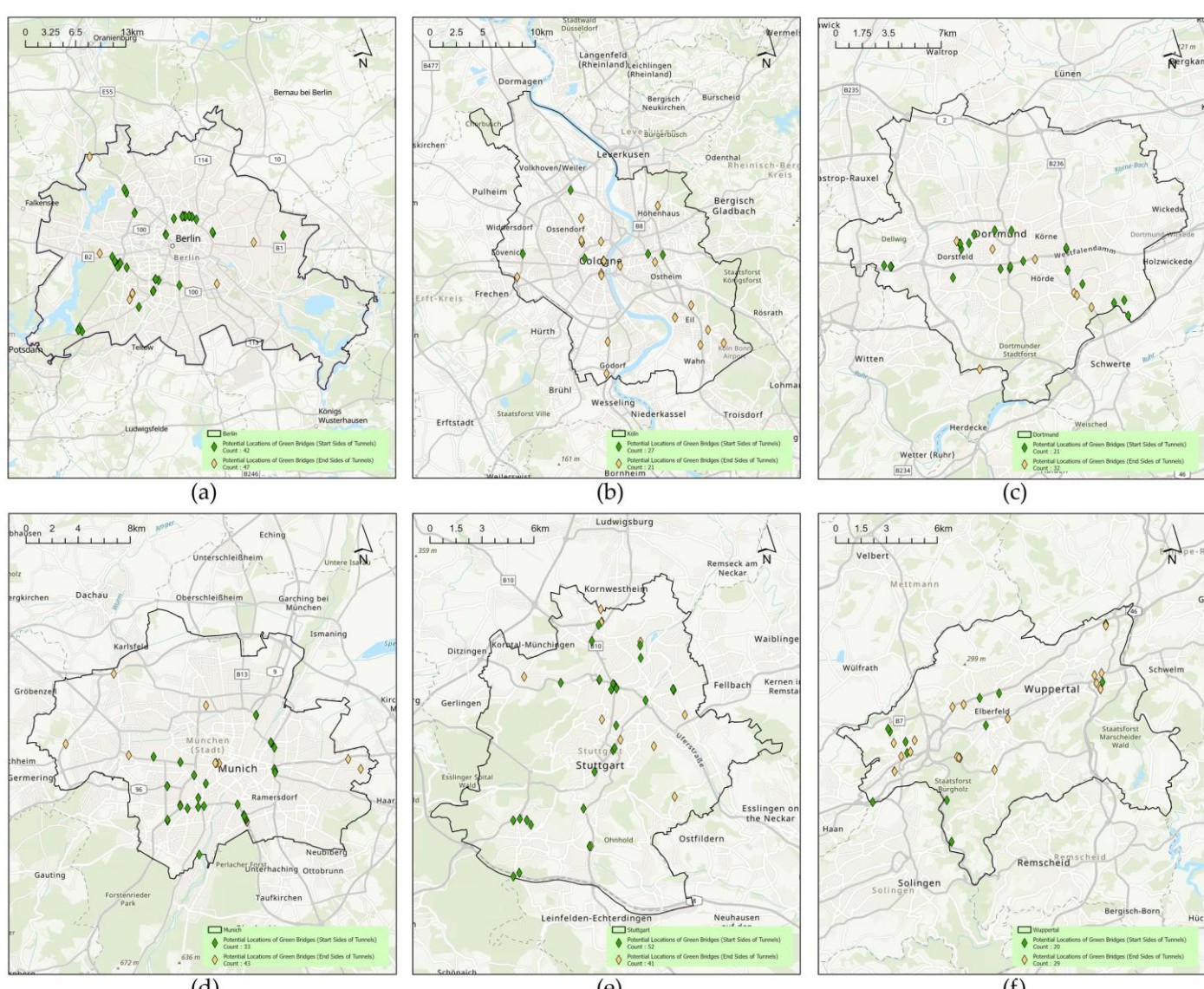

**Figure 8.** Cities with a relatively large number of geometrically feasible locations (**a**) Berlin, (**b**) Cologne, (**c**) Dortmund, (**d**) Munich, (**e**) Stuttgart, (**f**) Wuppertal.

Figure 9 provides detailed representations of some individual locations. These locations were confirmed after validating them with Google Earth Pro. Results show that the potential location of a green bridge is likely to be near the existing green bridges (Figure 9a,c,g). The extraction of potential locations is based on the 100 m distance from a tunnel. However, it does not mean that the potential locations are only at that point. For example, Figure 9c,f shows that the potential location may extend to the other side of the

tunnel. Moreover, Figure 9d,e shows that many potential locations can be concentrated in the same area which allows the realization of a longer green bridge.

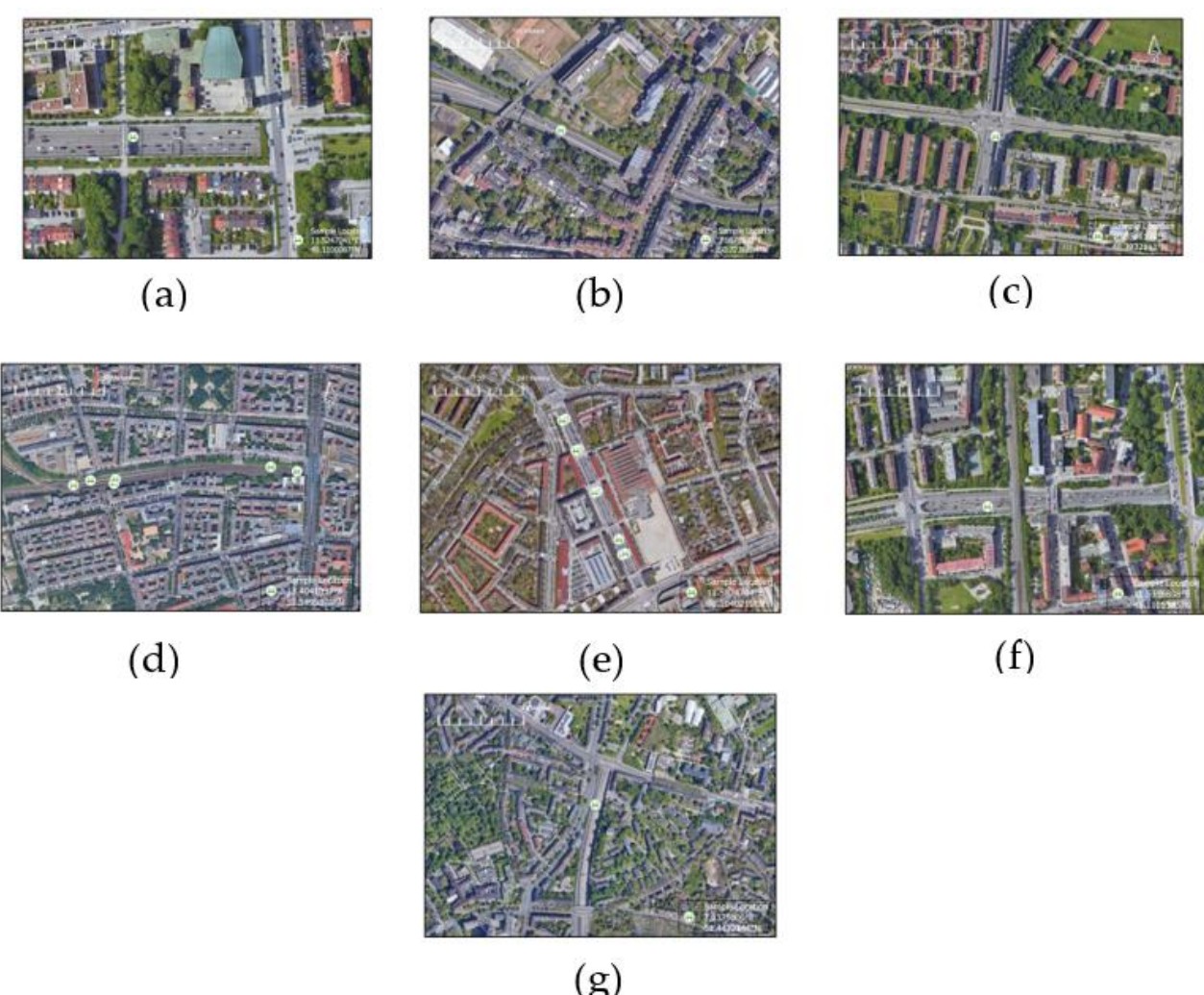

**Figure 9.** Detailed maps of individual potential locations (**a**) E54 147-135, Grünstraße, 81369 Munich, (**b**) Reuterstraße, 53115 Bonn, (**c**) B300, 86156 Augsburg (**d**) Malmöer Str. 29, 10439 BerlI(**e**) McGraw Trench, Munich (**f**) Sendling-Westpark Munich (**g**) Huttrop, 45138 Essen.

Results show that Germany has relatively high coverage of existing green spaces at the national level (Figure 10a). The air quality map shows northern Germany has a higher concentration of PM2.5 than the southern part, especially on the west and east sides (Figure 10b). Residential areas follow major cities, which are spatially similar to the population density distribution to some degree (Figure 10c,n). Industrial areas seem to be more spatially related to the intensity of urban heat island areas (Figure 10d). Average monthly income is high in major cities, such as Munich, especially in the southern part (Figure 10e). Flood depth is remarkable in some small rivers, especially in the lower middle part while the flood extent is larger in the northern and southern part of the country (Figure 10g). Hospitals are concentrated in the north-western part and the southern part (Figure 10h). Majority of critical infrastructure such as schools, markets, kindergartens are found to be highly concentrated in the western part except for power stations which are mostly clustered in the south-eastern corner of the country (Figure 10k). In contrast, the elderly population is mainly clustered in the north-eastern side of Germany.

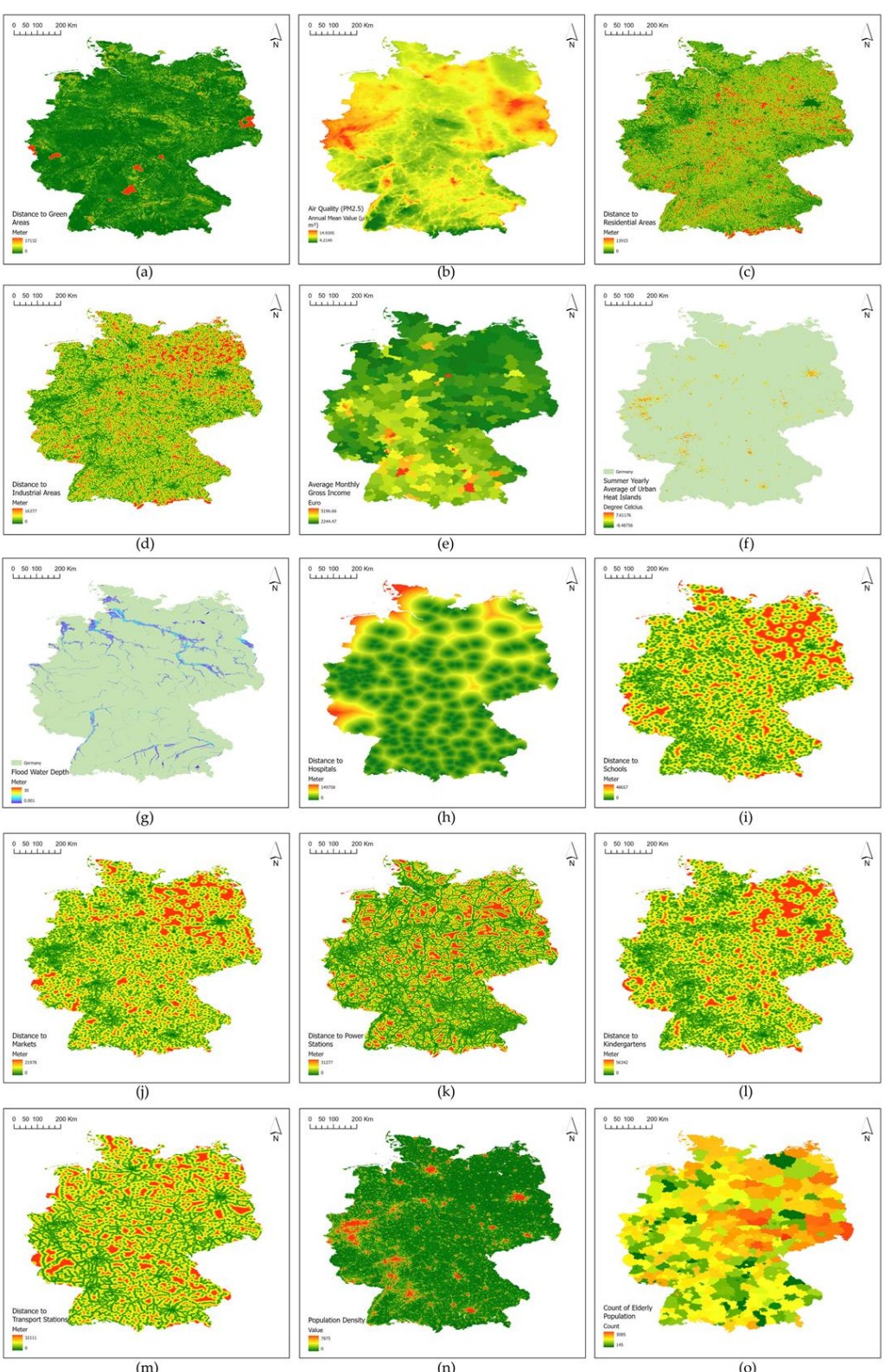

**Figure 10.** Individual spatial layers as each criterion used in the MCDA analysis: (**a**) extent of green space areas; (**b**) amount of air pollutants (PM2.5); (**c**) extent of residential areas; (**d**) extent of industrial aIs; (**e**) average monthly income; (**f**) intensity of urban heat island areas; (**g**) depth of flood hazard areas; (**h**) density of hospitals; (**i**) density of schools; (**j**) density of kindergartens; (**k**) density of markets; (**l**) density of power stations; (**m**) density of transport stations; (**n**) population density; (**o**) density of elderly people.

### 3.2. Multi-Criteria Analysis

The spatial suitability of green bridges follows spatial patterns of major cities. The distribution patterns are similar to where the population, critical infrastructure, urban heat island areas, residential and industrial, and high-income cities are concentrated. Suitable areas also cover major spots of air pollution. The relations between green space and the suitability map are not apparent at the provided zoom level in the National scale map. In other words, the detailed differences in the proximity (i.e., distance) of existing green areas and the suitable locations may be better reflected at the scale of individual locations.

Results from the expert survey (Table 1) show that existing green space has the highest importance for implementing a green bridge while the density of the power stations is the least weighted parameter. Infrastructure criteria are the ones with relatively less importance. Stress criteria such as heat, flood, and air pollution have relatively high weights. The combined weights of population, infrastructure, land use, and heat stress criteria seem to provide a high influence on the spatial patterns of the suitability map.

Figure 11b shows the suitable locations with their suitability classes. There are 287 locations in the highest suitability class with a suitability score of 6636 to 7176. This is followed by 220 locations in the second-highest suitability classes. The differences in the suitability scores between the classes are small. This means that the second and third classes also have high suitability to implement green bridges. The last three classes do not have green bridge locations because those classes are distributed on the outskirts of the cities.

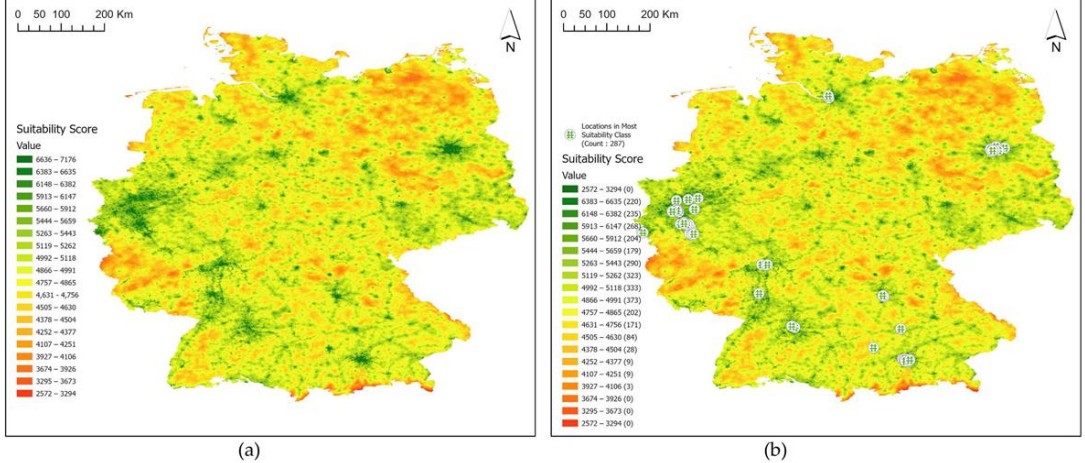

(a) (b)

**Figure 11.** (**a**) Suitability map with suitability score (highest suitability in green color) and (**b**) suitability map with potential locations in the highest suitability class. Number next to the score represents the number of locations included in each class.

Results show that some individual locations in the highest suitability class appear to be in areas with limited green spaces at this zoom level. These spatial results of suitable locations are in accordance with the relative importance of the weight for the green area criteria (Table 1). Locations in Figure 12a,b have comparatively high implementation potential as many suitable locations are concentrated in those spots. Implementation of the green bridge in those areas would produce longer green bridges and make their implementation more efficient.

Results show that some individual locations in the highest suitability class appear to be in areas with limited green spaces at this zoom level. These spatial results of suitable locations are in accordance with the relative importance of the weight for the green area criteria (Table 1). Locations in Figure 12a,b have comparatively high implementation potential as many suitable locations are concentrated in those spots. Implementation of a green bridge in those areas would produce longer green bridges and make their implementation more efficient.

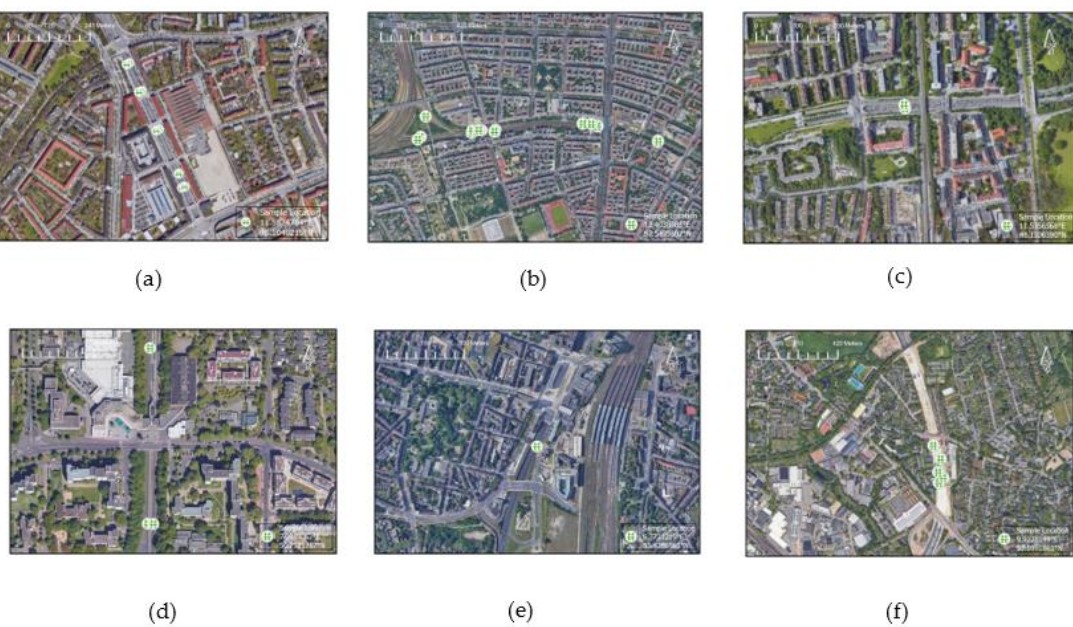

**Figure 12.** Sample locations from the highest suitability class (**a**) McGraw Trench, Munich (**b**) Malmöer Str. 29, 10439 Berlin (**c**) Sendling-Westpark Munich (**d**) Tannenbusch Mitte, Oppelner Str., 53119 Bonn (**e**) A59, 47051 Duisburg (**f**) A7, 22527 Hamburg.

The existence of some street trees might compromise location suitability as cutting those existing trees to implement a green bridge is not considered appropriate (Figure 12d). However, a nearby already existing green bridge is preferable as it can enhance the connectivity functions of the green areas. Moreover, the location in Figure 12b is in front of a major train station in Berlin, which can improve the accessibility of green space and property buildings to the majority of users. Most importantly, the suitable locations in Figure 12f from Hamburg are at the exact spot where the Humburg Deckel project is implementing a green bridge project what is called "Humburg roofs" [1,29]. This is real-ground proof of the validity of the geometry analysis toolbox and MCDA of this paper. As the Humburg Deckel project is still under construction, the spot was captured in our analysis.

Figure 12a shows the McGraw Trench in Munich, which is a case study that we used for a detailed cost-benefit analysis. As this location was suitable in geometry analysis and MCDA, it is further analyzed for its costs and benefits to understand its economic feasibility (See Section 3.3).

### 3.3. Cost and Benefit Analysis

A case study in Munich is used here as a representative case for the cost and benefit analysis. Table 6 shows that the coverage area has a size of 9000 m$^2$, which is divided into seven buildable modules (M2) and 13 green bridge modules (M1) according to the specified BCR. The total area is 13,500 m$^2$. The GFA for MFH of the project corresponds to the maximum permissible FAR and amounts to 21,600 m$^2$, which represents a usable area of 14,407 m$^2$. In addition, the usable floor space of the basement is 3881 m$^2$. Basement rooms of 909 m$^2$ and 84 parking spaces can be built in the basement. A detailed table with the determined values (masses, as well as costs and revenues) can be found in Supplementary Materials.

**Table 6.** Distances and areas of masses in case study of McGraw Trench.

| Parameters | Determined Values |
| --- | --- |
| Length project | 450 m |
| Width (standard cross section) | 20 m |
| Cover surface area | 9000 m$^2$ |
| Total floor area | 13,500 m$^2$ |
| Development area basement | 6405 m$^2$ |
| Basement usable area | 3881 m$^2$ |
| Parking and traffic area | 2972 m$^2$ |
| Cellar rooms | 909 m$^2$ |
| Parking spaces | 84 |
| Used BCR | 0.3 |
| Used FAR | 1.6 |
| Permissible BCR | 4050 m$^2$ |
| Permissible FAR | 21,600 m$^2$ |
| SFH | 0% |
| MFH | 100% |
| GFA MFH project | 21,000 m$^2$ |
| Project usable area | 14,407 m$^2$ |
| Quantity M1 | 13 |
| Total area M1 | 3960 m$^2$ |
| Quantity M2 | 7 |
| Total area M2 | 5040 m$^2$ |
| Quantity M3 | 0 |
| Total area M3 | 0 m$^2$ |

### 3.3.1. Costs

The total costs and the cost breakdown of the case study are shown in Table 7. The total 21pprox.of approx. €116.2 million are made up of the construction costs of the cover with ca. €73.8 million, the development of the basement with ca. €4.3 million, and the development with ca. €38.1 million.

**Table 7.** Total cost of McGraw trench (cost status 3Q 2022; incl. VAT.).

| Cost Group | Designation | McGraw Trench |
| --- | --- | --- |
| 1. Cover | | 73,800,550 € |
| CG 200 | Preparatory measures/terrain modelling | 2,300,671 € |
| CG 300 | Building-cover construction, modules | 51,419,968 € |
| | M1 | 12,221,235 € |
| | M2 | 39,198,733 € |
| | M3 | 0 € |
| CG 400 | Technical installations tunnel | 16,784,195 € |
| CG 500 | Outdoor area next to cover (OA1) | 2,494,706 € |
| CG 500 | Outdoor area on cover, without M1 (OA2) | 801,010 € |
| 2. Basement extension | | 4,310,370 € |
| | Extension cover (basement) | 4,310,370 € |
| 3. Building development | | 38,083,099 € |
| CG 300–400 | SFH | 0 € |
| CG 300–400 | MFH | 38,083,099 € |
| | Total | 116,194,019 € |

3.3.2. Benefits

In the case study, a GFA of 21,600 m$^2$ is calculated, as already listed under "Areas". With a share of 66.7% of the GFA, this corresponds to a useable area of 14,407 m$^2$. In addition, revenues can also be generated from 84 underground parking spaces, which will be built.

The costs of the cover per square meter of total area amount to 5467 €/m$^2$, which can be equated with the land costs. A comparison of the estimated standard land value and the costs of the cover shows that the assumed standard land value of 2900 €/m$^2$ is slightly more than half of the calculated land costs.

The costs of the development per square meter of living space, on the other hand, amount to 2643 €/m$^2$. Apportioning the construction costs of the cover incl. the basement (total costs) to the living space results in a total price per square meter of 8065 €/m$^2$ (incl. basement). In order for the project to be economically viable, these costs must be covered by the income. The sale prices determined from the property market report of the internet portal Immoscout24 result in an average sale price of flats in the Munich district of Obergiesing of approximately 8850 €/m$^2$ (Table 4). The costs of the bridge could therefore be covered by the sale price of the property alone.

Table 8 shows the monetary benefits of the green bridge in the case study. Results show that cultural ecosystem services have the highest monetary value. However, this value is very subjective to the people. It is based on what benefits people receive from nature and opportunities such as recreation spots and education [49]. The value of cultural services is followed by temperature regulation and water retention values. The latter two services are important to protect people from the impacts of heat waves and floods. Their relative importance can also be seen in the expert weights of climate impact criteria (Table 1). Air pollution and carbon sequestration services have the lowest values. The total benefits of the green bridge through selling the property and annual soft benefits (only for a single year) heavily outweigh the total costs of the green bridge. Even though the sale price of the property alone could outweigh the costs, the intangible benefits of the green bridge should not be neglected. It is important to note that the return value of selling the property is for one time, the monetary benefits of the green space will be received annually forever. Results also show that annual monetary benefits in the renting scenario can compensate for the total cost after 38 years.

**Table 8.** Monetary benefits of green space (M1 and M2 module) in the green bridge of the case study.

| Benefits | Monetary Value |
| --- | --- |
| Benefits of water retention | 546.73 €/year |
| Air pollution control | 287.47 €/year |
| Carbon sequestration | 206.92 €/year |
| Temperature regulation | 451.7 €/year |
| Cultural ecosystem services | 1946.25 €/year |
| Total benefits of green space | 3442.87 €/year |
| Sale price of M2 | 8850 €/m$^2$ |
| Rental price of M2 | 208.68 €/m$^2$/year |
| Total monetary income for selling M2 | 127,501,950 € |
| Total rental price of M2 | 3,006,452.76 €/year |
| Total benefits for the green bridge (through renting property module (M2)) | 3,009,895.63 €/year |
| Total benefits for the green bridge (through selling property module (M2)) | 127,505,392.9 € |

Results of sensitivity analysis (Figure 13) show that BCR and FAR have the highest influence on the overall costs per m². The higher the BCR and FAR values are, the lower the cost is. Therefore, dense development represents the best case. The worst case here is clearly the loose development.

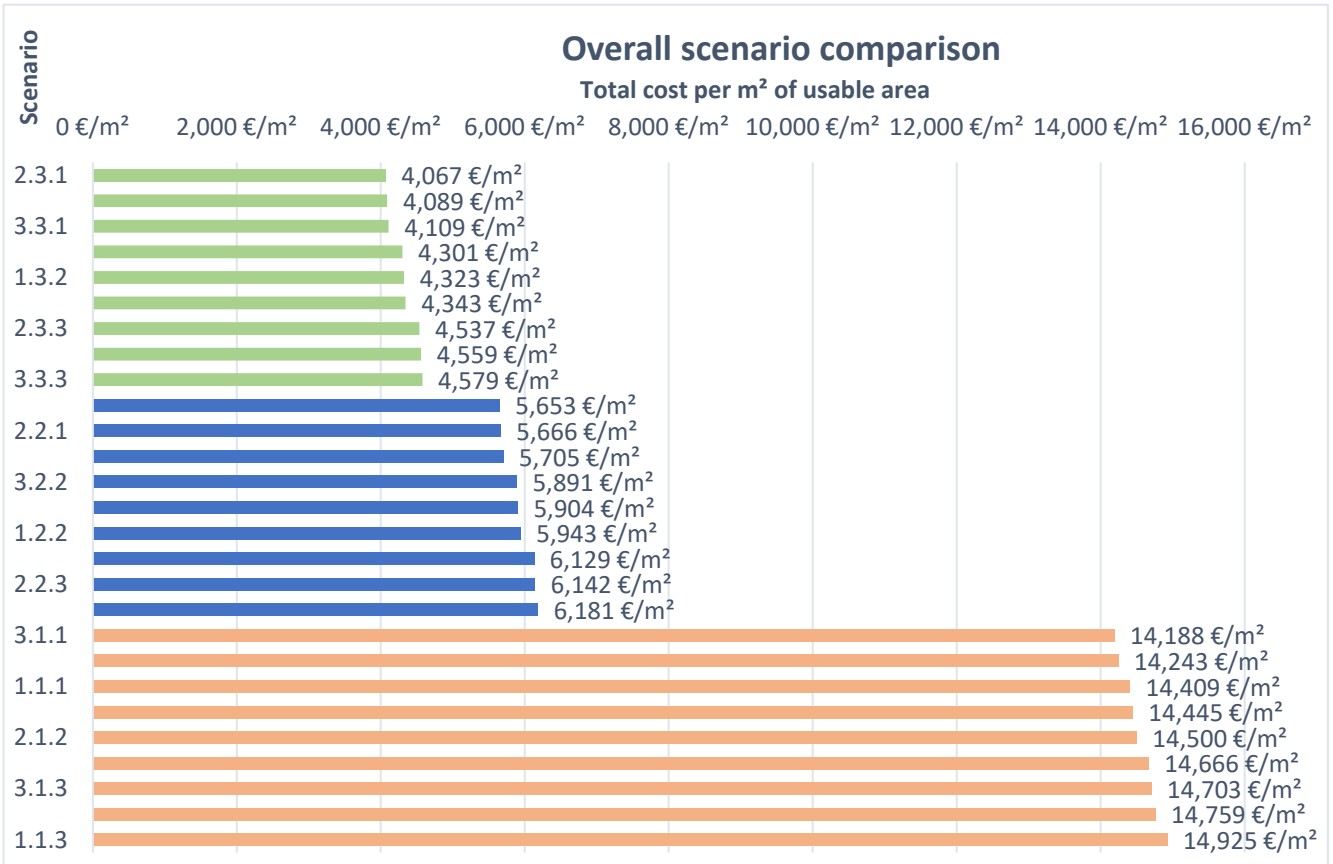

**Figure 13.** Costs per different pairs of scenarios (sensitivity analysis).

## 4. Discussion

This paper presents a pioneering assessment of the potential for implementing green bridges at a national scale, with a focus on identifying practical and suitable locations. It provides the spatial implementation potential of green bridges in three major procedural analyses. These analyses reflect the three research objectives of the paper. These objectives are aligned with the work packages of the associated SuLiVaCo project.

The first objective, which is the identification of geometrical feasibility using the geometry analysis toolbox is the most important analysis that determines the suitable locations for green bridges in practice. It is also an objective criterion that is fundamental for a green bridge location selection. The automatic function of the toolbox at the spatial resolution of 5 m can support upscaling of the initiative to other countries. Since the geometry analysis is related to the roads and tunnels in urban areas, the data, especially the elevation data, should have adequately high resolution.

However, the toolbox also has limitations. Due to the limited processing capacity, the road datasets for the whole of Germany were grouped into seven parts and processed. As a result, the roads were disconnected according to the boundaries of counties. Therefore, virtual extensions of tunnels were stopped in the periphery boundary areas of cities. This results in a false selection of green bridge potential locations on the tunnels (instead of near the tunnels) without the virtual extension in the peripheral area of cities. However, this limitation is assumed to be negligible as our major focus is on the inner-cities areas and their traffic lanes.

The results of MCDA were greatly influenced by the weights assigned. The low importance of critical infrastructure is probably because most experts may assume infrastructure criteria for the purpose of accessibility to that critical infrastructure. While the purpose of accessibility is still valid, the hidden intention of including those criteria, was the protection of those critical infrastructures using services of green spaces. Normally, the collapse or the damage of that critical infrastructure, for example, due to extreme heat, floods, or air pollution, may have the greatest impact on the wider population since they are the centers of critical services.

Some of the selected locations (Figure 12) are near the existing green bridges, which is positive in the aspect of connectivity for a larger green space. However, this connectivity factor can conflict with the criteria of existing green space as a negative factor, which has the purpose to identify areas in need of green spaces. In addition to connecting the green space, the green bridge that we defined here also has functions to connect neighborhoods by creating new spaces. Since we did not use the width of a road as a parameter or a criterion, the results are mixed with narrow roads or railways and larger roads in the MCDA. It is recommended to use this criterion to classify wider green bridge locations that enhance the suitability quality in the future. Another recommended criterion is to consider the existence of street trees in the potential locations to be implemented since the removal of the trees for the implementation of a green bridge is not considered appropriate. Another limitation is that land values and building price criteria were not included in the MCDA due to the limited availability of those datasets.

The suitability map seems to follow the spatial patterns of major cities in Germany. This happens because many criteria such as heat impact, critical infrastructure, population density, and residential and industrial areas have the same patterns. This allows to select relatively high number of locations (i.e., 287 locations) in the highest suitable classes. It is also in accordance with the focus of the paper to implement green bridges in inner-city areas. The number of locations that coincide with the highest suitable classes also depends on the size of the classes. For example, the score can be categorized into 10 classes which results in a higher number of suitable locations and may allow more alternative selections.

While the geometry feasibility and cost-effectiveness are objective factors to determine green bridge locations and implementation potential in reality, most of the criteria used in the MCDA are subjective and reflect varied opinions among interdisciplinary experts. This is especially obvious in the weighting of the criteria. The different opinions reflected in defining the purpose of a green bridge have a high influence on determining the importance of the criteria. Therefore, it is reasonable that we split the former two analyses from the MCDA. This type of bias preference may be reduced by having a large number of expert samples from a diversity of fields to achieve holistic consensus.

In addition to the differences in preferences and opinions, another important consideration in the MCDA is the definition of each criterion to make a spatial layer. The fuzzy definitions can result in different outcomes. For example, our definition of green spaces includes community allotments and recreation grounds [50]. A universal definition of green spaces should define its categories.

The sensitivity analysis of the cost model proved that dense development has the least cost impact. This is also true when sharing a green space as a private property and a public property, where the latter allows access to a community and creates a sense of cohesion. Here, the dense development can favor the use of a larger public green space. Moreover, if BCR and FAR affect the cost, it should also use the width of a road as an additional criterion to provide higher suitability in the suitability analysis for their actual implementation. Moreover, the width of a green bridge can provide additional insights than consideration of the length of a potential green bridge.

Regarding the legal feasibility of green bridges in Germany, there are laws and policies to support the implementation of green infrastructure like green bridges. For example, Germany's urban green infrastructure policy provides opportunities for the establishment of green infrastructure like a green bridge [51]. Moreover, another important potential legal

basis is Federal Nature Conservation Act (BNatSchG), which supports nature conservation actions [52]. Proper interpretations to make regarding legal aspects of green bridges are out of our scope and further research can focus on proper legal and policy analysis to ensure the strong legal feasibility of green bridges.

As the cost-benefit analysis proved the economic feasibility of the green bridge case study, it is highly recommended to implement green bridges at locations in the high suitability classes (at least three highest suitable classes). The use of light-weight structures may be one of the reasons of the reduced costs of a green bridge [28]. Moreover, the monetary benefits of a green bridge, from selling the residential property alone, can outweigh the cost of its implementation. While comparing the results of costs and benefits from a feasibility study in Freiberg, the selling prices were higher in the case study which provides more economic benefits due to the limited living space in the city of Munich [28]. As a limitation, it should include distributional differences of the ecosystem services in the calculation of monetary benefits of green spaces [53,54].

## 5. Conclusions

The concept of the lightweight urban green bridge can enhance the sustainability and resilience of cities because it creates new spaces for green spaces and residential areas. The use of lightweight structures also reduces the costs of masses effectively. This paper provides important implications for the practical implementation of green bridges in reality and opportunities to upscale to other countries. The geometry toolbox which identifies potentially suitable locations for light green bridges can help to an upscale larger scale. The geometry and elevation considerations were specialized for the feasibility to implement a green bridge. The MCDA also provides scientifically sound results to further classify potential locations according to their suitability using properly adjusted weight factors set by experts. We found 3249 geometrically feasible locations and 287 most suitable locations according to MCDA. It is important to bear in mind that the inclusion of additional parameters such as "width of roads" can provide greater insights for their implementation ability. The cost-benefit analysis proves the economic feasibility of green bridge implementation. The ecosystem services that are included in the calculation of monetary value provide not only monetary benefits but also several social and ecological co-benefits. Connectivity aspects, enhanced by lightweight green bridges, should be considered an added value to the areas where the bridges are implemented.

**Supplementary Materials:** The following supporting information can be downloaded at: https://www.mdpi.com/article/10.3390/rs15030753/s1, S1: OpenStreetMap Tags used for some criteria used in the Multi-criteria Decision Analysis The OSM tags define each criteria. Column names are keys and the body text were their values.; S2: Detailed description of area calculation of the modules and their arrangement; S3: Cost calculations for cost parameters.

**Author Contributions:** Conceptualization, M.B., J.H. (Johannes Hawlik) and M.H.; Methodology, H.W.Y.K., J.H. (Jocelyne Hellwig), M.B., J.H. (Jocelyne Hellwig) and M.H.; Software, H.W.Y.K. and J.H. (Johannes Hawlik); Validation, H.W.Y.K., J.H. (Jocelyne Hellwig), J.H. (Johannes Hawlik) and M.H.; Formal analysis, H.W.Y.K. and J.H. (Jocelyne Hellwig); Investigation, J.H. (Jocelyne Hellwig), M.B., J.H. (Johannes Hawlik) and M.H.; Resources, H.W.Y.K., J.H. (Jocelyne Hellwig), M.B., J.H. (Johannes Hawlik) and M.H.; Data curation, H.W.Y.K., J.H. (Jocelyne Hellwig) and J.H. (Johannes Hawlik); Writing–original draft, H.W.Y.K.; Writing–review & editing, A.C., J.H. (Jocelyne Hellwig), M.B., J.H. (Johannes Hawlik) and M.H.; Visualization, H.W.Y.K. and J.H. (Jocelyne Hellwig); Supervision, M.B., J.H. (Johannes Hawlik) and M.H.; Project administration, M.B., J.H. (Johannes Hawlik) and M.H.; Funding acquisition, M.B., J.H. (Johannes Hawlik) and M.H. All authors have read and agreed to the published version of the manuscript.

**Funding:** This research was funded by Ministry of Science, Research and the Arts Baden-Württemberg with grant number of Az. 32-7533-4-113.0/25/3 And The APC was funded by MDPI Remote Sensing Journal.

**Data Availability Statement:** The dataset regarding geometrically feasible locations of green bridges and locations within the first five suitability classes as a result of Multi-criteria Decision-Making Analysis for potential green bridges in Germany can be found in the following link: https://drive.google.com/file/d/1tVMbsbk7LShrj3YNyRRSjoyCG_ugG3KW/view?usp=sharing, accessed on 3 January 2023. Geometrically feasible locations present potential green bridge locations that are geometrically possible to implement in terms of height and length of a green bridge. The data is the result of geometry toolbox. Suitability Classes present most suitable green bridge locations among geometrically feasible locations based on multiple factors as a result of Multi-Criteria Decision-Making Analysis. Five most suitability classes are provided.

**Conflicts of Interest:** The authors declare no conflict of interest.

**Acknowledgments:** We express our sincere thanks to Phone Pyai Tun for his great technical support. We are also genuinely thankful to Patrick Scheib, Paula Meyer, Christian Süss, and Kai Schwarz for their great support with the data and content of the paper.

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
