# Peer review of "Multifactorial Evaluation of Spatial Suitability and Economic Viability of Light Green Bridges Using Remote Sensing Data and Spatial Urban Planning Criteria"

_remotesensing, doi:10.3390/rs15030753_

Round 1

Reviewer 1 Report

The paper deals with a very interesting topic, but the structure of the paper needs to be improved before publication. Here are some aspects that could be revised:

(1) Remote Sensing journal required that “The abstract should be written as one paragraph (of about 300 words).”. Please revise it accordingly.

(2) What does “Nature-based Solutions (NbS)” refers to? What kind of technologies are involved? What is the aim of nature-based solutions? These must be explained in the Introduction (before addressing the need for NbS).

(3) Introduction is very loosely connected: The author use five paragraphs to introduce the barriers of NbS, but what is NbS and why NbS is needed is unclear. The financial arrangement of NbS and the feasibility test are introduced, but the connection of the literature to this paper is vague. Adding some transition sentences between each paragraph can be helpful.

(4) I suggest the authors move the introduction of NbS and green bridge to the beginning of the introduction and then focus the introduction on 1. Why green bridge is important for urban ecology? 2. What are the key challenges in green bridge implementation? And how geometry analysis can help bridge the gap? 3. What are the current analyses on the green bridge? And how is your paper different from theirs?

(5) Research questions can be better rephrased with question marks. At present, it is more of a statement of what you did, rather than what question you have addressed.

(6) Section 2.2.2: the summary statistics of experts need to be disclosed (education, years of experience, field of expertise) 

(7) Conclusion: I suggest the author add more specific key findings from your paper. At present, the conclusions are too general.

(8) Minor issues: 

    1. Line 420 and 422, the number appears in the middle of sentences "151 and 24".
    2. English writing does not have grammar issues, but the flow of sentences could be improved.

Author Response

Dear Reviewer,

Thank you so much for your thoughtful comments. We really appreciate your comments and the time you spent for our paper. Please kindly find our response below in the attachment. 

Warm regards,

Hnin Wuit Yee Kyaw

Reviewer 2 Report

The article Multifactorial evaluation of spatial suitability and economic viability of light green bridg

The article Multifactorial evaluation of spatial suitability and economic viability of light green bridges using remote sensing data and spatial urban planning criteria

The abstract is too long. Please rewrite it respecting the requirements of the journal both in terms of length and structure. Lines 17-21 as well as 26-30 can be deleted from the abstract and moved, if desired, to the introduction section.

I appreciate the goal of creating a toolbox for identifying locations of green bridges as a space creation mechanism considering geometry and elevation of road networks of cities using an automated GIS-based toolbox.

We also appreciate the involvement of engineering firms for the design of green bridges.

Figure 2. Processes of Geometry Analysis Toolbox. It should be scrapped. In the current format it is difficult to read, If you remove the background and write with a more suitable font, it would be better to understand what you want to convey.

In line 167, specify that OpenStreetMap data has been classified into 7 sections. Which are these?

The work is truly a technique, in some places difficult to follow, but it has great practical applicability and it is appreciable that the analysis was carried out for the entire surface of Germany.

For Figure 4. cities with a relatively large number of geometrically feasible locations (a)Berlin, (b) Cologne, (c) Dortmond, (d) Munich, (e) Stuttgart, (f) Wuppertal. the label for each locality is positioned on the center of the maps, so they cover the information on the maps. Move the names of the localities to the corner, up.

The maps from Figure 6. Individual spatial layers as each criteria used in the MCDA analysis (a) Extent of Green Space areas (b) Amount of air pollutants (PM2.5) (c) Extent of Residential Areas (d) Extent of Industrial Areas (e) Average monthly income (f) Intensity of Urban Heat Islands Areas (g) Depth of flood hazard areas (h) Density of hospitals (i) Density of schools (j) Density of kinder-gartens (k) Density of markets (l) Density of power stations (m) Density of transport stations (n) Population Density (o) Density of elderly people. They do not have a legend that can be read. You should also add some identification elements, a polygon with the study area, some main cities, etc.

For Figure 7. (a) Suitability Map with suitability score (highest suitability in green color) and (b) suitability map with potential locations in the most suitability class (right) Number next to the score represents the number of locations falls in each class.please detele value from the legend of the maps

Add a discussion about legal aspects to the discussion section. What are the limitations in terms of laws, local in Germany but also at the level of the European Union for the creation of these types of green bridges.

Why did you added this part before the Bibliography section??

Publication bibliography

Dymek, Dominika; Wilkaniec, Agnieszka; Bednorz, Leszek; Szczepańska, Magdalena (2021): Significance of Allotment Gardens in Urban Green Space Systems and Their Classification for Spatial Planning Purposes: A Case Study of Poznań, Poland. In Sustainability 13 (19), p. 11044. DOI: 10.3390/su131911044. 766

Herrmann, Michael; Arnold, Mark; Fentzloff, Arne; Otterbach, Simon; Hawlik, Johannes; Kunz-Wedler, Lilly (2020): Abschlussbericht FREIRAUM für FREIBERG: Eine Landschaftsbrücke in hybrider Leichtbauweise mit Wohn- und Bürobebauung über die A81 für mehr Grün in Freibergs Mitte.

Ministry of Urban Development and Housing Humburg (2020): SPACIOUS AND QUIET NEW GREEN SPACES ON TOP OF THE A 7. Available online at 771

https://www.hamburg.de/contentblob/14644794/d0ae8dfb950d7625f7f67f91193a1745/data/d-broschuere-%E2%80%9Ea7-deckel-englisch-letzter-stand%E2%80%9C.pdf, checked on 10/31/2022.

Road and Transportation Research Association (2018): Guidelines for the Design of Motorways. Available onlineat https://www.fgsv-verlag.de/pub/media/pdf/202_E_PDF.v.pdf, checked on 10/14/2022.Robinson, Lisa A.; Hammitt, James K.; Adler, Matthew D. (2018): Assessing the Distribution of Impacts in Global Benefit‐Cost Analysis. In SSRN Journal. DOI: 10.2139/ssrn.4014003.

es using remote sensing data and spatial urban planning criteria

The abstract is too long. Please rewrite it respecting the requirements of the journal both in terms of length and structure. Lines 17-21 as well as 26-30 can be deleted from the abstract and moved, if desired, to the introduction section.

I appreciate the goal of creating a toolbox for identifying locations of green bridges as a space creation mechanism considering geometry and elevation of road networks of cities using an automated GIS-based toolbox.

We also appreciate the involvement of engineering firms for the design of green bridges.

Figure 2. Processes of Geometry Analysis Toolbox. It should be scrapped. In the current format it is difficult to read, If you remove the background and write with a more suitable font, it would be better to understand what you want to convey.

In line 167, specify that OpenStreetMap data has been classified into 7 sections. Which are these?

The work is truly a technique, in some places difficult to follow, but it has great practical applicability and it is appreciable that the analysis was carried out for the entire surface of Germany.

For Figure 4. cities with a relatively large number of geometrically feasible locations (a)Berlin, (b) Cologne, (c) Dortmond, (d) Munich, (e) Stuttgart, (f) Wuppertal. the label for each locality is positioned on the center of the maps, so they cover the information on the maps. Move the names of the localities to the corner, up.

The maps from Figure 6. Individual spatial layers as each criteria used in the MCDA analysis (a) Extent of Green Space areas (b) Amount of air pollutants (PM2.5) (c) Extent of Residential Areas (d) Extent of Industrial Areas (e) Average monthly income (f) Intensity of Urban Heat Islands Areas (g) Depth of flood hazard areas (h) Density of hospitals (i) Density of schools (j) Density of kinder-gartens (k) Density of markets (l) Density of power stations (m) Density of transport stations (n) Population Density (o) Density of elderly people. They do not have a legend that can be read. You should also add some identification elements, a polygon with the study area, some main cities, etc.

For Figure 7. (a) Suitability Map with suitability score (highest suitability in green color) and (b) suitability map with potential locations in the most suitability class (right) Number next to the score represents the number of locations falls in each class.please detele value from the legend of the maps

Add a discussion about legal aspects to the discussion section. What are the limitations in terms of laws, local in Germany but also at the level of the European Union for the creation of these types of green bridges.

Why did you added this part before the Bibliography section??

Author Response

Dear Reviewer,

Thank you so much for your thoughtful comments and feedback. We really appreciate your careful comments and the time you spent for our paper. Please kindly find our response in the attachment.

Warm regards,

Hnin Wuit Yee Kyaw

Round 2

Reviewer 1 Report

1. The references are not properly numbered (e.g., second paragraph: [43] - [45])

2. Issue I raised for point 3 still needs to be addressed: 

Can Nature-based solutions mitigate the impact such as floods and air pollution? Is there any reference to support that?

When introducing Nature-based solutions, when and why is this concept being proposed? Perhaps cite the definition from the European Commission? 

"On account of multifunctional benefits of Nature-based solutions, they are proliferating as an integrated approach which contribute to achieving to all the Sustainable Development Goals (SDGs) [44]." How does this Nature-based solution contribute to SDG 4 (Education)?

The financial arrangement of NbS and the feasibility test are introduced, but the connection of the literature to this paper is vague. Adding some transition sentences between each paragraph can be helpful.

Author Response

Dear Reviewer, 

Many thanks for your kind comments for the improvement of the paper. We really appreciate your careful consideration and review. Please kindly find the attached answers to your good comments. If you are not still satisfied enought, please feel feel to let us know to go for another round! 

Many thanks and warm regards,

Hnin Wuit Yee Kyaw on behalf of author team
